# Preliminary Clinical Surgical Experience with Temporary Simultaneous Use of an Endoscope during Exoscopic Neurosurgery: An Observational Study

**DOI:** 10.3390/jcm11071753

**Published:** 2022-03-22

**Authors:** Yasuo Murai, Kazutaka Shirokane, Shun Sato, Tadashi Higuchi, Asami Kubota, Tomohiro Ozeki, Fumihiro Matano, Kazuma Sasakai, Fumio Yamaguchi, Akio Morita

**Affiliations:** 1Department of Neurological Surgery, Nippon Medical School, 1-1-5 Sendagi, Bunkyo-ku, Tokyo 113-8602, Japan; kazutaka-shirokane@nms.ac.jp (K.S.); s3049@nms.ac.jp (S.S.); higu2525@nms.ac.jp (T.H.); ak0813@nms.ac.jp (A.K.); t-ozeki@nms.ac.jp (T.O.); s00-078@nms.ac.jp (F.M.); k-sasaki@nms.ac.jp (K.S.); amor-tky@umin.ac.jp (A.M.); 2Department of Neurosurgery for Community Health, Nippon Medical School, 1-1-5 Sendagi, Bunkyo-ku, Tokyo 113-8602, Japan; fyamaguc@nms.ac.jp

**Keywords:** endoscope, exoscope, microscope, microsurgery, Japan, observational study

## Abstract

The use of an endoscope in exoscopic transcranial neurosurgery for skull-base lesions has not yet been investigated. Thus, this study aimed to investigate the advantages, disadvantages, and safety of “simultaneous temporary use of an endoscope during exoscopic surgery” (exo-endoscopic surgery (EES)). Consecutive exo-endoscopic surgeries performed by experienced neurosurgeons and assistants were analyzed. Surgical complications and time were compared with previous consecutive microsurgeries performed by the same surgeon. A questionnaire survey was conducted on 16 neurosurgeons with experience in both “temporary simultaneous use of endoscope during microscopic surgery” (micro-endoscopic surgery (MES)) and EES. EES was performed in 18 of 76 exoscopic surgeries, including tumor removal (*n* = 10), aneurysm clipping (*n* = 5), and others (*n* = 3). There were no significant differences in operative time, anesthesia time, or complications from microsurgery by the same operator. According to the questionnaire survey results, compared with MES, EES had a wider field of view due to its lack of an eyepiece, was easier when loading and unloading instruments into and out of the surgical field, and was more suitable for the simultaneous observation of two fields of view. Overall, 79.2% of surgeons indicated that EES may be better suited than MES to simultaneously observe two fields of view.

## 1. Introduction

With the advancement of digital image processing technology, the miniaturization of video equipment, and the introduction of high-resolution digital imaging equipment, such as 4 K, 8 K, and three-dimensional (3D) screens to the medical field, there have been increasing reports on the usefulness and safety of exoscopes and heads-up surgery in the field of neurosurgery [1,2,3,4]. Various advantages have been pointed out for heads-up surgery, which is microsurgery without a microscope (MS) but with monitored observation using a high-resolution exoscope. These advantages included the surgeon’s posture, allowing him to stay away from the eyepiece, and the absence of the eyepiece, resulting in an enlarged field of view for the surgeon. Previous reports also indicate that heads-up surgery improves surgical safety by improving ergonomics, such as the surgeon’s posture and field of view expansion [1,3,5]. However, exoscopes have not been demonstrated to be more useful than a MS in deep and especially narrow fields of view [1,3,6,7]. In addition, exoscopes have been suggested to be inferior to a MS in transsphenoidal surgery and intraventricular tumors [1,6,8].

Microsurgery in the field of neurosurgery has been performed since the 1950s. The usefulness and limitations of the “temporary simultaneous use of an endoscope during microscopic surgery” (herein, micro-endoscopic surgery (MES)) were not reported until the 1990s; it has been used for deep and back-surface observation of anatomical objects [9,10,11,12,13]. For example, endoscope-assisted observation of the perforating branch behind an aneurysm, which helps obtain the necessary information for clipping that cannot be observed with a MS alone, is possible [10]. In addition, recent high-resolution technological advances in rigid MS and advances in rigid MS fixtures have improved their functions as tools to support microscopic surgery [14,15]. However, the following problems have been suggested to be disadvantages of MES: the possibility of damage to the surrounding tissues when the endoscope is inserted and removed due to the limitation of the field of view; endoscope interference with instrument insertion, such as scissors and tweezers; and the complexity of alternating between the fields of view of the MS and the endoscope [10,11,16]. When a surgeon is operating using an MS, it is difficult to directly observe the surgical field because the eyepiece and objective lens of the MS are in the way, thus limiting the optimal view of the surgical field. However, in exoscopic surgery, the surgeon can observe both the magnified surgical field on the monitor and the surgical field at hand almost simultaneously with only minor eye movements, which contributes to the safety of the surgery. The authors observed that the exoscope has a wider field of view for the surgeon than that of the MS [1,17], which may be useful for the insertion and removal of rigid endoscopes and surgical instruments into the operative field.

Previous studies have reported the usefulness of switching to endoscopes when the field of observation is inadequate in exoscopic surgery [6,8,17]. However, we searched for studies on the “simultaneous use of endoscopes and exoscopes” but did not find any. In addition, “the simultaneous temporary use of an endoscope during exoscopic surgery” (herein, exo-endoscopic surgery [EES]) and MES have not yet been compared. Therefore, the purpose of this study was to investigate the potential contribution of EES to surgical safety and simultaneous visualization of the two fields of view in comparison to MES.

## 2. Materials and Methods

### 2.1. Study Design

This observational study aimed to compare the usefulness and safety of EES and MES by examining the following two aspects: (1) comparison of operative time, anesthesia time, and frequency of complications associated with EES and MES performed by the same surgeon for the same period; and (2) questionnaire survey administered to neurosurgeons who had participated in both EES and MES as surgeons or assistants regarding their comparison. One primary surgeon (Y.M.) was in charge of all cases of cerebral aneurysms, whether exoscopic or microscopic. Another primary surgeon (A.M.) was in charge of all cases of auditory nerve tumors, whether endoscopic or microscopic. For surgeries using the exoscope, written consent for the procedure was obtained from all patients using a form approved by the institutional review board of Nippon Medical School Hospital.

### 2.2. Instruments

We used the exoscope to replace the surgical MS in hospitals and prepared both the surgical MS (Carl Zeiss Pentero 900, Carl Zeiss Meditec AG, Jena, Germany) and an exoscope for exoscopic surgery. The following three types of endoscopes were used: ORBEYE (OLYMPUS, Tokyo, Japan), VITOM 3D (VITOM^®^, Karl Storz, Tuttlingen, Germany), and Kinevo 900 (Carl Zeiss Meditec AG, Jena, Germany). Between July 2018 and March 2019, 76 surgical cases were performed using an exoscope.

EES was performed in 18 patients. All surgeons were unfamiliar with the use of the exoscope; therefore, it was continued only when, at the discretion of each surgeon, it could be safely continued. The decision to switch to a surgical MS was made when using the exoscope was deemed impossible. When the exoscope and endoscope were used together, the operation was performed by either observing two different scope screens on a 55-inch monitor with two systems of input (Figure 1, video) or placing two 28-inch monitors side by side.

At our hospital, in conventional MES, the endoscope screen is observed using a 28-inch ceiling-suspended monitor, and the microsurgical surgeon moves his gaze from the eyepiece to the screen. Of the 18 cases of combined endoscopic procedures, 17 were performed with ORBEYE and one with Vitom 3D. In addition, there were two cases involving the combined use of the Kinevo 900 and the QUEVO endoscope; however, these were excluded from this study because they entailed observations only.

The same surgical instruments, such as bipolar forceps and micro scissors, were used in all surgeries for both MES and EES. Olympus rigid endoscopes (0°, 30°, 70°; outer diameter, 2.7 mm; length, 18 m) and an EndoArm endoscopic holder (Olympus, Tokyo, Japan) were used in all EES. In both exoscopic and microscopic surgeries, the use of an endoscope was left to the surgeon’s discretion.

### 2.3. Methods

The 18 cases included vestibular schwannoma removal (*n* = 6), anterior circulation cerebral aneurysm clipping (*n* = 5), meningioma removal (*n* = 4), transcranial removal of craniopharyngioma (*n* = 1), transcranial removal of the epidermoid (*n* = 1), and microvascular decompression for trigeminal neuralgia (*n* = 1) (Appendix A). Vestibular schwannoma removal and cerebral aneurysm clipping were compared with consecutive cases performed by the same surgeon from 2017 to 2020 in terms of postoperative complications, operative time, and anesthesia time. One of the two neurosurgeons had performed more than 800 cases of surgery for vestibular nerve tumors as a primary surgeon, whereas the other had performed more than 500 cases of craniotomy for cerebral aneurysms as a primary surgeon. These two neurosurgeons served as either the primary surgeon or the first assistant surgeon for all 18 cases. The other 14 surgeons took turns as first to third assistant. Eight board certified neurosurgeons served as the primary surgeons or first assistants in all 18 surgeries. Additionally, only four surgeons were responsible for EES among all 18 surgeries. Therefore, the number of times these 14 surgeons washed their hands and participated in surgery ranged from 3 to 12 times. In addition, they observed the surgeries.

All cases of vestibular schwannoma were operated on via the lateral suboccipital approach using EES, and the tumor size was <28 mm. The cases used for comparison were consecutive cases with the same conditions and performed by the same surgeon; tumor sizes were <28 mm. In addition, cerebral aneurysm clipping using an exoscope was <17 mm in all cases, and the cases included frontotemporal craniotomy without concomitant revascularization. The cases used for comparison were also consecutive cases performed by the same surgeon under the same conditions. These case groups were compared in terms of postoperative complications, operative time, and anesthesia time.

All cases of cerebral aneurysm clipping by EES were initial operations for unruptured cerebral aneurysms in the anterior circulation. Therefore, only the initial clipping of unruptured cerebral aneurysms in the anterior circulation was selected for comparison. All cases of vestibular schwannoma removal by EES were also initial surgeries performed via the lateral suboccipital approach. Only microsurgical cases under the same conditions were included in the comparison. Patients with neurofibromatosis were excluded from the study. No selection was made based on patient age. Postoperative complications were assessed using the evaluation system used in Japan (National Council for University Hospital Medical Safety Management [Appendix A]). After all surgeries were completed, a questionnaire survey was administered to neurosurgeons who participated in endoscope-assisted microsurgeries as a surgeon or assistant surgeon and those who had participated in EES as a surgeon or assistant surgeon. 

### 2.4. Statistical Analyses

Non-normally distributed continuous variables were compared between groups using the Mann–Whitney U test. Categorical variables, such as the frequency of complications, were compared using Fisher’s exact test. Statistical significance was set at *p* < 0.05. All analyses were performed using JMP software (version 14.0; SAS Institute, Cary, NC, USA). The appropriate sample size and power were determined for the statistical analyses. Categorical variables were compared using Fisher’s exact test. The type 1 error was 5% (α = 0.05), the power was 95% (1 − β = 0.95), a *p*-value < 0.05 was considered statistically significant, and JMP 14.0 software was used for the analyses. Consequently, logistic regression analysis was excluded due to the small sample size. Responses to the questionnaire were rated on a five-point Likert scale as follows: strongly prefer MS, prefer MS, neither agree nor disagree, prefer exoscope, and strongly prefer exoscope. Each of these aspects was rated as follows: the grade “Neither agree nor disagree” denoted “Equal or Incapable” (equal to the MES or unable to judge), “Agree to EES” denoted “Superior”(EES is almost equal or superior to the MES), and “Strongly agree to EES” denoted “Clearly superior” (EES is clearly superior to the MES). We attempted to compare the seven questionnaire items with each other and extract the items that were more highly rated. Chi-square test was used for statistical analysis. Next, we divided the questionnaire evaluators into two groups: experienced board certified neurosurgeon and inexperienced non-board certified neurosurgeon, to investigate whether the evaluations differed depending on the experience of the surgeons. Fisher’s exact test was used for statistical analysis. 

## 3. Results

### Case Series

There were no significant differences in the operative time, anesthesia time, operative complications, or frequency of complications between the EES group or MES group for either aneurysmal clipping or acoustic tumor removal (Table 1). For vestibular schwannoma tumors, age was significantly lower in the microsurgery group (*p* = 0.018).

Details of the 11 cases of EES are shown in Table 1. All were transient complications; however, there was one case of vestibular schwannoma associated with dysphagia and one case of posterior fossa epidermoid tumor associated with hoarseness. Among the five patients who underwent aneurysmal clipping, no residual aneurysm was confirmed on postoperative 3D computed tomography angiography. The two surgeons who served as primary surgeons had performed more than 1000 microscopic surgeries and less than 30 exoscopic surgeries.

Although this study mainly aimed to evaluate EES, no cases were switched to microscopy while performing EES. In cases of cerebral aneurysm clipping (*n* = 1), epidermoid removal (*n* = 1), and microvascular decompression for trigeminal neuralgia (*n* = 1), the surgeons found it difficult to operate with the exoscope and switched to the MS owing to the narrow field of view at depth. However, in two of these cases, the exoscope was used in situations where temporary endoscopic use was necessary. 

Valid responses to the questionnaire survey were obtained from 16 neurosurgeons. The survey items and a summary of the results is provided in Table 2. Of the 112 valid responses regarding endoscopy-assisted surgery, 5.4% felt that the MS was superior, 15.2% felt that the MS and exoscope were equivalent, and 79.2% felt that the exoscope was superior. The questionnaire survey results indicated potential superiority of the EES to the MES in terms of the ease of viewing of the surgical instruments and the surgical monitor, and of the ability to move instruments in and out of the deep surgical field because of the wide field of view. There were no items for which the exoscope was evaluated as inferior to the MS (Table 2). Therefore, we examined whether there were any items among the seven evaluation items in which the EES was rated higher. As a result, in comparison with the ease of preparation, deep insertion of the endoscope (*p* = 0.035), simultaneous observation of the two monitors (*p* = 0.041), and observation of the surgeon’s hand (*p* = 0.035) were rated statistically significantly higher. The results are shown in Table 2. Subsequently, the physicians who participated in the questionnaire survey were divided into two groups: board-certified neurosurgeons (N = 8), who served as primary surgeons or first assistants, and non-board-certified neurosurgeons (N = 8), who served as assistants. The results of the EES and MES questionnaires (Table 3) were statistically examined between the two groups, but none of the items reached statistical significance.

## 4. Discussion

We want to emphasize that this study did not to aim to examine the superiority or inferiority of the visual fields of exoscopic surgery and microscopic surgery. Rather, this study aimed to determine whether the exoscope or the MS was superior when used in combination with an endoscope. The light is unidirectional for both the MS and exoscope, and the areas behind and beside the optical axis are difficult to see with either instrument. Thus, it has been recommended to temporarily use an endoscope simultaneously with an MS to expand the field of view. This study demonstrates that the exoscope may be superior to the MS in some aspects that are limited to the temporary use of the endoscope. This study consisted of a limited number of cases, and further investigation involving a larger number of cases is needed. It is also necessary to investigate this subject in terms of different surgical positions and pathologies.

The comparison between MES and EES revealed that there was no significant difference in operative time, anesthesia time, or frequency of complications, although surgeons had limited experience with this technique. The survey results indicated that the ease of insertion of the endoscope into the field of view, the simultaneous observation of the two surgical fields, and the ease of viewing the surgeon’s field of view at hand were highly rated. Furthermore, the survey results were not influenced by the level of experience of the neurosurgeons.

The surgeons felt that EES provided a wider field of view owing to the absence of eyepieces and the possibility of safely moving surgical instruments, such as bipolar forceps and micro-scissors, and the endoscope itself into and out of the surgical field [1,3,6]. The fundamental difference between MES and EES is that EES is a heads-up surgery in which the two fields of view are simultaneously monitored [5,6,10,13]. With some instruments, it is possible to have a picture of the endoscope screen in the MS’s field of view; however, in MES [18,19], the surgeon must look away from the MS eyepiece and watch the endoscope display while performing the operation. We concluded that MES has an advantage such that eye movements are almost unnecessary.

Micro-endoscopic surgery expands the field of view in deeper regions of the body [9,10,11]; however, various issues have been pointed out [10,16,18], including the difficulty in simultaneously observing both the endoscopic and microscopic fields of view, complexity of setting up the monitoring equipment in the operating room [12,18], interference of the endoscope itself and endoscope fixation devices with surgical operations [12,16], and tissue damage during the insertion and removal of the equipment [10]. A search on PubMed showed that many studies on the advantages of MES have been published since 1998. The number of reports on the application of the exoscope in neurosurgery increased in 2018 [1,17]. However, as of December 2021, we could not identify any papers on MES. We hypothesize that problems associated with surgeons needing to move their eyes can be resolved by EES.

### 4.1. Questionnaire Survey Results

All the responses were obtained from neurosurgeons with little experience in exoscopic surgery. However, when analyses were limited to endoscope-assisted surgery, we found that 5.4% of the responses favored the MS, and 79.2% favored the exoscope. However, it should be noted that this result does not indicate that the exoscope is superior to the MS in all surgeries. This study entailed only a comparison between the exoscope and the MS in surgeries using an endoscope. We also observed that the MS was rated relatively higher in terms of ease of preparation in the operating room. This may be partly attributed to the limited experience of the surgeons with exoscopic surgery using an endoscope.

We decided to include all assistants in the questionnaire survey instead of limiting it to only the surgeon, the reason being that one of the characteristics of exoscopic surgery is that participants other than the surgeon can observe the surgical field in exactly the same way as the surgeon [1,3]. This has been pointed out to be useful both from the perspective of surgical education and surgical safety. We thought it was necessary to examine whether this kind of evaluation of exoscopic surgery could also be applied in endoscopy-assisted surgery.

### 4.2. Future Research Directions 

3D images are becoming increasingly available for both exoscopic [6,7,20] and endoscopic surgery screens. At our hospital, when using ORBEYE, heads-up surgery is performed with a single-screen 3D display on a 55-inch monitor for observation using only an exoscope, whereas endoscope-assisted ORBEYE surgery is performed with a double-screen display on a 55-inch monitor. However, in this study, a 3D display for endoscope-assisted ORBEYE surgery was not possible. To the best of our knowledge, as of the spring of 2021, there are no monitors available on the market that can display both screens in 3D, even if they are capable of 3D observation. For this reason, although 3D images are becoming increasingly available for exoscopic and endoscopic surgeries, it is necessary to install two monitors that can project 3D images side by side to permit observation of both scopes in 3D. It is also necessary to have the same 3D system that can support the same 3D glasses for both; otherwise, the 3D glasses will need to be replaced.

The survey on the usefulness of this surgery is based on a questionnaire survey. The survey should ideally be based on objective figures, not only on the impressions of the surgeon or assistant. Therefore, as a way of increasing objectivity, it is desirable in the future to classify a large number of cases in different surgical positions and pathologies, and to investigate the operative time and complications.

### 4.3. Limitations

This study had some limitations. First, this is only an initial report, and the number of cases in this study is limited. We searched PubMed for reports on preliminary studies on exoscopic neurosurgery published since 2019 using the search terms “exoscope,” “microsurgery,” and “neurosurgery”. We found seven studies [1,21,22,23,24,25,26] with a mean number of cases of 21.4 (range, 18–29). Moreover, the number of EES included in this study is smaller than that of general exoscopic surgery. Based on these results, we concluded that the number of cases in this study was not too small for preliminary study. In this study, we set 6 months as the study period for the initial experience, conducted a questionnaire survey, and reported the survey results to the participants. Therefore, we did not add any cases after this period because that would have resulted in different results from those obtained in the initial experience. Second, because the results of this study evaluated MES, there were two procedures performed beyond the study period in which the decision was made to switch to exoscopic surgery during microsurgery as simultaneous endoscopic assistance was deemed necessary. However, it is also necessary to compare the results when more surgeries are performed using an exoscope. We noted that the surgeon’s anterior field of view was wider when using the exoscope; however, we could not evaluate the difference quantitatively. In microsurgery, the anterior visual field is not zero, even when the surgeon looks through the eyepiece. Lastly, we included Japanese patients only. Future studies should increase the number of cases analyzed and determine the contribution of surgery to safety using multivariate regression methods.

## 5. Conclusions

We report our initial experience with EES. MES and EES had similar operative times and complications. Results of the survey showed that more than 79% of surgeons reported a possible advantage of EES over MES, and the simplicity of preparing for EES was rated low. In the future, the development of a large monitor, capable of simultaneously displaying two screens of 3D images, is anticipated.

## Figures and Tables

**Figure 1 jcm-11-01753-f001:**
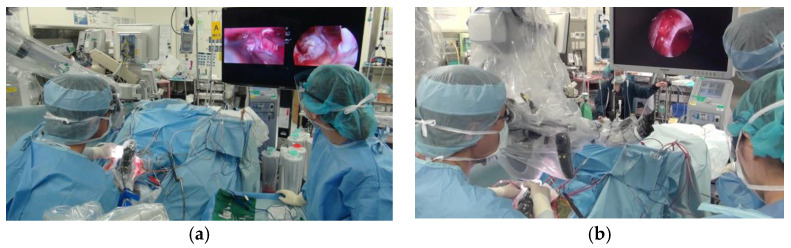
Scenes in the operating room during EES and MES. (**a**) A scene in the operating room during EES for craniopharyngioma through the right peritoneal approach in the supine position showing the endoscope and the exoscope on two screens of a 55-inch monitor. The surgeon has an unobstructed view of the two camera screens and can observe them simultaneously. The assisting nurse can easily observe the surgeon’s hand and the monitor. (**b**) A typical view of MES. The surgeon looks away from the MS eyepiece and looks only at the endoscope monitor. The surgeon cannot even see his hands. Neither the assisting nurse nor the assistant surgeon can see the microscopic field of view.

**Table 1 jcm-11-01753-t001:** Comparison of microscopic surgery and exoscopic surgery.

	Microscopic Surgery	Exoscopic Surgery	*p* Value
Vestibular schwannoma			
*n*	36	6	
Male:female ratio	20:16	1:05	0.092
Age, years, median (IQR)	49 (17.5)	69 (12.75)	0.0018
Surgical time (m), median (IQR)	375.5 (128.75)	352 (112.75)	0.843
Anesthesia time (m), median (IQR)	509 (115.75)	484 (114)	0.788
Complications (number of cases)	8	1	0.756
Aneurysm clipping			
*n*	60	5	
Male:female ratio	24:36	2:3	0.671
Age (years), median (IQR)	61.5 (17.75)	60 (16.5)	0.596
Surgical time (m), median (IQR)	260.5 (117)	311 (67.5)	0.273
Anesthesia time (m), median (IQR)	358.5 (120.75)	402 (100.5)	0.172
Complications (number of cases)	1	0	0.771

IQR, interquartile range; m, minutes.

**Table 2 jcm-11-01753-t002:** The five-point Likert scale evaluation by neurosurgeons (N = 16).

		Strongly Agree to Micro-Endoscopic Surgery	Agree to Micro-Endoscopic Surgery	Neither Agree nor Disagree	Agree to Exo-Endoscopic Surgery	Strongly Agree to Exo-Endoscopic Surgery	Statistical Results with Item 1
1	Which method do you think is more convenient to prepare for?	0	4	3	4	5	
2	In which method is it easier and safer to insert the endoscope into the deep surgical field?	0	0	1	8	7	0.035
3	In which method is it easier and safer to insert and withdraw the suction tube or bipolar forceps in and out of the surgical field?	1	0	5	4	6	0.327
4	In which method is it easier for the surgeon to monitor the endoscopic field of view?	0	1	2	3	10	0.165
5	In which method is it easier for the surgeon to monitor the endoscopic field of view as well as the microscope or the exoscope simultaneously?	0	0	2	2	12	0.041
6	In which method is it easier for the nurse or assistant surgeon to monitor simultaneously the microscopic or exoscopic field of view in addition to the endoscopic field of view?	0	0	3	5	8	0.096
7	Which method is more likely to allow the assisting surgeon or nurse to see the surgeon’s hand and the surgical field?	0	0	1	9	6	0.035

**Table 3 jcm-11-01753-t003:** Comparison of survey results based on the level of neurosurgeon experience (N = 16).

		Board-Certified Neurosurgeons Who Prefer the Exoscope (N = 8)	Non-Board-Certified Neurosurgeons Who Prefer the Exoscope (N = 8)	*p* Value
1	Which method do you think is more convenient to prepare for?	6	3	0.157
2	In which method is it easier and safer to insert the endoscope into the deep surgical field?	7	8	0.500
3	In which method is it easier and safer to insert and withdraw the suction tube or bipolar forceps in and out of the surgical field?	5	5	0.695
4	In which method is it easier for the surgeon to monitor the endoscopic field of view?	7	7	0.766
5	In which method is it easier for the surgeon to monitor the endoscopic field of view as well as the microscope or the exoscope simultaneously?	7	8	0.500
6	In which method is it easier for the nurse or assistant surgeon to monitor simultaneously the microscopic or exoscopic field of view in addition to the endoscopic field of view?	7	6	0.500
7	Which method is more likely to allow the assisting surgeon or nurse to see the surgeon’s hand and the surgical field?	8	7	0.500

## Data Availability

The data presented in this study are available on request from the corresponding author.

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
