# Peer review of "Preliminary Clinical Surgical Experience with Temporary Simultaneous Use of an Endoscope during Exoscopic Neurosurgery: An Observational Study"

_jcm, 2022, doi:10.3390/jcm11071753_

Round 1

Reviewer 1 Report

Response to the authors

The objective of this clinical study (with 18 patients) was to evaluate the contribution of exo-endoscopic surgery to surgical safety in comparison to micro-endoscopic surgery. For that purpose, the authors analyzed surgical complications and time were compared with previous consecutive microsurgeries performed by the same two surgeons. Additionally, they performed a survey (with 7 items) on 16 surgeons who were involved in these cases. In summary, they present no significant differences in operative time, anesthesia time, or complications between exo-endoscopic and micro-endoscopic surgeries. These results are based (in contrast to the complete sample size) on 11 exo-endoscopic cases (6 vestibular schwannoma and 5 aneurysm clippings) compared to 96 micro-endoscopic cases. The case selection was supposed comparability concerning the surgeons. The survey showed that approximately 80% of the surgeons felt that exo-endoscopic surgeries were superior to micro-endoscopic surgeries in terms of the ease of viewing of the surgical instruments and the surgical monitor, and of the ability to move instruments in and out of the deep surgical field because of the wide field of view. The authors conclude that this studies has proven superiority of exo-endoscopic surgeries.

The report is well-written in an adequate English language. The introduction provides a good overview of the available literature and outlines the problem that should be addressed by the study.

The legibility is impeded by the continuous repetition of the complex terms “exo-endoscopic”, “micro-endoscopic” and “microsurgery”. The reviewer suggests to use abbreviations. However, there are major drawbacks of the study.

  • It remains unclear how many surgeons in fact performed the surgeries. The second assistant is usually not the one performing the surgery. This is important as exo-endoscopic surgeries will demonstrate a learning curve. This fact is not addressed by the authors.
  • In fact, this study enrolls only 11 prospective patients. Thus, the sampling size is really small to allow final conclusions.
  • There is no single statistics on the results of the questionnaire.

Thus, the study is finally not of sufficient quality for publication in the current version.

Author Response

Revision Letter to Reviewer 1:

Thank you very much for your insightful remarks, which helped improve the content of the paper.

We have corrected the abbreviations for "exo-endoscopic surgery (EES)," "micro-endoscopic surgery (MES)," and "microscope (MS)" as you pointed out.

I have added detailed information about the surgeons in the Materials and Methods section. One primary surgeon is in charge of the surgery of all cerebral aneurysms and another one for all acoustic tumors, whether microscopic or exoscopic.

As you suggested, we have added a description of the statistical study. The discussion of the paper has been improved by your suggestions.

From the questionnaire survey, the EES was rated higher than the MES. Therefore, we examined whether there were any items among the seven evaluation items in which the EES was rated higher. We found that although the evaluation of the ease of preparation was relatively low, the evaluation of two items, namely, deep insertion of the endoscope and simultaneous observation of two surgical fields, was high.

Next, we also divided the physicians involved in the questionnaire survey into two groups: board-certified neurosurgeons who served as primary surgeons or first assistants, and non-board-certified neurosurgeons who served as assistants.

The results (new table 3) of the EES and MES questionnaires were statistically examined between the two groups, but none of the items reached statistical significance.

We also included in the Discussion section the reasons why we included all assistants in the questionnaire survey. In particular, it has been pointed out that exoscopic surgery is characterized by the fact that participants other than the surgeon can observe the surgical field exactly the same as the surgeon. This is because all members of the team can share information during the operation, and it has been reported that this is useful from the viewpoint of surgical education; thus, it is necessary to confirm the usefulness of this feature in exoscopic surgery with an endoscope.

Reviewer 2 Report

In this paper, the authors report their preliminary experience with simultaneous use of endoscope during exoscopic surgery and compare this experience with their huge experience with micro-endoscopic surgery.
The topic is extremely interesting and the authors should be commended for their effort. Indeed the team is very strong, uses up-to-date technology and has no fear of innovation.
The two cohorts are almost equivalent in terms of operative or anesthesia time and complications.
However, the second cohort of endoscope-assisted exoscopic surgery consisted only of 18 cases.
It is true it is clearly stated among limitations, but I think a so small group is not sufficient to “determine whether the exoscope or the microscope was superior when used in combination with an endoscope” as declared at the beginning of the discussion. One should also take into account that these 18 cases were operated on for different pathologies and in different positions, which clearly could influence time, complications, surgeons' feelings about safety and efficacy of the combined procedures.

Author Response

Thank you very much for your helpful comments that helped improve our report. We have added the points you raised to the study limitations.

In the Future research directions section, we added a note that a more detailed study is needed in terms of operative time, complications, safety, and efficacy by classifying the different pathologies and positions.

Reviewer 3 Report

In this study the the authors aimed to investigate the advantages, disadvantages, and the safety of exo-endoscopic surgery. Surgery was performed in 18 of 76 exoscopic surgeries (Tumor removal n=10; aneurysm clipping N=5; and others N=3).

They examined two aspects: 1. Operative time, anesthesia time, and frequency of complications associated with exo-endoscopic surgery and micro-endoscopic surgery performed by the same surgeon at the same periode were compared; and 2. Questionnaire survery administered to neurosurgeons who had participated in both exo-endoscopic surgery and micro-endoscopic surgery as surgeons or assistants regarding there comparison.

They found , that here is no significant difference in operative Time, anesthesia time or frequency of complications.

They conclude, that the exoscopic surgery can improve the simultaneous visualization of the operative field in neurosurgery, which requires the use of an endoscope and increased the safety during surgery.

I am not

convinced by this manuscript.

The authors report preliminary results of a technique, which is wildely used sine 2017 and of which 55 papers are published in the literature. Furthermore it is a single center trial in which only 18 patients were included.

The questionnaire they used in there study is a“self-made“ test of 7 questions in relation to the abovementioned two procedures. Only 16 surgeons or assistens participated in this questionnaire. Therefore no conclusions can be drawn from these results.

In section „Results“ the authors describe that there was no significant difference in operative Time, anesthesia time or frequency of complications between both procedures: the exo-endoscopic surgery and micro-endoscopic surgery. Nevertheless the authors conclude that the exoscopic surgery can improve the simultaneous visualization oft he operative field in neurosurgery, which requires the use of an endoscope and increased the safety during surgery.

Also this conclusion cannot be drawn out of there study.  

Author Response

Revision Letter to Reviewer 3:

Thank you very much for your detailed review. Your remarks have helped us realize two important issues regarding our report.

The first is that the paper is overly worded, and the second is the lack of explanation of the originality of this report.

First, the results reported in this paper consisted of only a few cases and were obtained via a questionnaire survey. Although similar preliminarily reports have been published with the same number of cases as this report, the results should not be stated categorically, and it should be clearly emphasized that a larger number of cases should be investigated. We have added this information to the Abstract and Discussion section.

Next, we would like to discuss the originality of this paper. As the reviewers pointed out, there are many papers on the exoscope alone that have already been published. However, to our knowledge, there is no report on the significance or usefulness of the simultaneous use of an exoscope and an endoscope. Microsurgery in the field of neurosurgery has been performed since the 1950s, but the usefulness of endoscopy combined with microsurgery has only recently been reported in the 1990s, and new studies are still being reported to this day [10,11]. For this reason, we believe that our paper offers a different informational value from those of the many papers on exoscope alone. We have added a note on this point in the Introduction and Discussion section.

Round 2

Reviewer 1 Report

Thank for providing a revised version of the manuscript. I appreciate that you have statistically reanalyzed the questionnaire results. However, the aspect of responsible surgeons is not addressed. As far as I understand the manuscript there was one responsible senior surgeon but the performing surgeon could have changed. Thus, it reduces the OR per surgeons to a level not allowing a learning curve. 

Author Response

Thank you very much for your detailed review.

We have added sentences regarding the responsible surgeons (P3, L133-136).

Reviewer 2 Report

Ok

Author Response

Thank you very much for your detailed review.

Reviewer 3 Report

Unfortunately, my remarkes were not answered.

Author Response

We apologize for misunderstanding the aforementioned remarks. We have removed and completely rewritten the conclusion section, as suggested (P9, L319-323). The conclusion section of the abstract was also rewritten (P1, L24-28).